# Melatonin Administration Enhances Testicular Volume, Testicular Blood Flow, Semen Parameters and Antioxidant Status during the Non-Breeding Season in Bafra Rams

**DOI:** 10.3390/ani14030442

**Published:** 2024-01-29

**Authors:** Melih Akar, Mesut Çevik, Adem Kocaman, Cumali Kaya, Burcu Esin, Stefan Björkman

**Affiliations:** 1Department of Production Animal Medicine, Faculty of Veterinary Medicine, University of Helsinki, 00014 Helsinki, Finland; stefan.bjorkman@helsinki.fi; 2Department of Animal Reproduction and Artificial Insemination, University of Ondokuz Mayis, Samsun 55200, Türkiye; cevikm@omu.edu.tr (M.Ç.); cumali.kaya@omu.edu.tr (C.K.); burcuyalcin@omu.edu.tr (B.E.); 3Department of Histology and Embryology, Medical Faculty, University of Ondokuz Mayis, Samsun 55200, Türkiye; adem.kocaman@omu.edu.tr

**Keywords:** Bafra rams, melatonin, testicular morphometry, sheep, testis blood flow, sperm quality

## Abstract

**Simple Summary:**

Our study aimed to understand how melatonin affects reproductive parameters in Bafra rams during the non-breeding season. One group of rams received melatonin implants, while another served as the control. We measured testicular volume, blood flow, and semen quality over several weeks. The results showed that melatonin increased testicular size and blood flow. This suggests that providing melatonin implants before the non-breeding season may improve reproductive capacity.

**Abstract:**

Our objectives were to investigate the effects of exogenous melatonin on testicular volume (TV), testicular blood flow (TBF), and semen quality in Bafra rams during the non-breeding season. One group of rams (MEL, *n* = 5) received a 36 mg melatonin implant twice, with 30 days in between, while the other group (CON, *n* = 5) served as the control. TBF, TV, and semen quality parameters were determined at three-week intervals starting three weeks before until twelve weeks after the first melatonin implant. Testicular blood flow was determined in the supratesticular (STA) and marginal testicular artery (MA) using color Doppler ultrasound. Semen was collected and evaluated, and the total oxidative status (TOS) and total antioxidative status (TAS) was determined using an ELISA. The MEL group had increased (*p* < 0.05) TV between the sixth and twelfth week after the start of treatment. Overall, the MEL group had lower resistance and pulsatility indexes (*p* < 0.05) between the third and ninth week, although there was no difference (*p* > 0.05) between the two groups in most semen quality parameters. However, TAS concentrations increased (*p* < 0.05) in the MEL group compared with the CON. The results of this study show that exogenous melatonin in the non-breeding season significantly increased both TBF and TV in Bafra rams. Therefore, giving rams implants with 36 mg melatonin twice at least one month prior to the non-breeding season is expected to improve testicular size and function and reproductive capacity.

## 1. Introduction

The cyclical variation in daylight, together with several other elements like temperature, nutrition, interaction with other males, lambing time, and nursing duration, are the primary determinants of seasonality. Rams can also exhibit these seasonal variations in reproduction in intensive breeding, although to a lesser extent. In comparison to spring and summer, rams have better reproductive efficiency and semen quality during the breeding season (winter–autumn, northern hemisphere) [1]. Nevertheless, rams continue to reproduce and engage in sexual behavior even outside the breeding season [2].

The common factor connecting photoperiod, seasonality, and reproduction is melatonin, the pineal hormone (N-acetyl-5-methoxytrypamine) [3]. In seasonally breeding animals such as certain breeds of sheep and goats, melatonin has a major role in the regulation of reproductive functions in response to changes in the duration of daylight by the direct action on the hypothalamic–pituitary–gonadal axis [4]. The melatonin secretion pattern transmits information about the light/dark cycles to the physiological centers of the body to ensure the organization of seasonal and circadian rhythms [2]. In addition, it is a lipophilic agent that crosses lipid bilayers and acts as a free radical scavenger, neutralizing hydroxyl and peroxyl radicals, preventing lipid membrane peroxidation and apoptosis, and protecting DNA from damage caused by free radicals. Melatonin is also considered a potent biological antioxidant in the body and stimulates gene expression of antioxidative enzymes, including superoxide dismutase, glutathione peroxidase, catalase, and glutathione reductase [5].

Reproduction has a critical economic aspect in the livestock industry, which depends on both the female and male animals’ reproductive capacities. Regular monitoring of the reproductive ability of rams in a flock contributes to the early diagnosis of disorders and high-quality ram selection [6]. Knowing the morphometric testes dimensions and sperm characteristics in determining the fertility of rams has an important place in the creation of high-yielding breeds. Testicular size and sperm characteristics are important for ram fertility and rams with larger testes produce more sperm, and sperm number and quality also affect reproductive performance [7]. Morphometric evaluation of testes in rams includes evaluation of shape, size, and testicular echotexture [4]. Numerous studies on rams show the importance of testicular ultrasound in evaluating the correlation with sperm parameters in pubertal and mature animals [8]. Studies have also revealed the importance of testicular blood flow (TBF) in the diagnosis of various reproductive disorders and in evaluating the reproductive potential of males. There is little information on normal reference values for the effect of different seasons on testicular blood flow in farm animals [9]. Color Doppler ultrasonography can help to assess the TBF in testicular arteries and to include suitable rams in breeding programs [4]. Additionally, a few Doppler parameters, including the testicular artery’s resistive index (RI) and pulsatility index (PI), are thought to be potential indicators of seminal quality [10]. Our objectives were to determine effects of exogenous melatonin in rams during the non-breeding season on TBF, testicular volume, and semen quality and oxidant status.

## 2. Materials and Methods

### 2.1. Animals and Ethical Declaration

In this study, we used ten sexually mature male Bafra rams (*Ovis aries*). The rams were 58.7 ± 5.31 months old and kept in groups in a semi-intensive system. They received the same care and feeding throughout the study. This study was conducted at the University of Ondokuz Mayis in Samsun/Türkiye (northern latitude: 41.37°, eastern longitude: 36.21°) during the non-breeding season (from April to June). Rams received a diet containing 400 g of hay cubes twice a day. Mineralized salt licks and clean tap water were available ad libitum. Before the study, the health of each ram was checked, and their testes were examined with ultrasound along with general and specific examinations for anomalies or dysfunction. Variations in daylight periods during the study (from April 2022 to June 2022) ranged from 13 to 15 h. All animals had a good libido and were clinically healthy before starting the study. Measurement of testicular volume, monitoring of TBF, and semen collection were performed three weeks before (W −3), at the time of (W0), and every three consecutive weeks for twelve weeks (W3, W6, W9, W12) after the first melatonin administration. Ondokuz Mayis University Animal Experiments Local Ethics Committee provided an ethics report for this study (E-68489742-604.01.03-51893)

### 2.2. Melatonin Administration

The rams were randomly divided into two groups: melatonin group (MEL, n = 5) and control group (CON, n = 5). All MEL rams received implants with 36 mg melatonin (Regulin^®^, Ceva, Samsun, Türkiye) subcutaneously twice, at W0 and 30 days after the first administration. An assistant restrained the ram and held the ear in place. The skin of the outer ear was cleaned and disinfected with an antiseptic solution, and the implant was placed with an applicator under the skin into an area with minimal vascularization and skin–cartilage connection (Figure 1A).

### 2.3. Measurement of Testicular Volume

Foreign material and fleece on the scrotum were removed and the scrotum was washed and dried. Length, width, and thickness of each testis were determined with a digital caliper. The scrotal width was measured at the widest circumference without exerting pressure on the testis. The dorsoventral length of the right and left testes was measured without considering the caput and cauda parts of the epididymis (Figure 1B).
$$a=\frac{4}{3}\times \pi \times \frac{b-k}{2}\times {\left[\frac{c-k}{2}\right]}^{2}$$

The craniocaudal thickness of the right and left testes was measured over the scrotum considering a correction factor (*k*) for the scrotal thickness. The testicular volume was calculated according to Milczewski et al. [11], i.e., with a basic ellipsoid equation where *a* = testicular volume (cm^3^), *b* = length of testis (cm), *c* = width of testis (cm), and *k* = scrotum thickness (cm).

### 2.4. Ultrasonographic Examinations

All examinations were performed using a color Doppler (Esaote Biomedica MYLAB5 VET, Shenzhen, China) with a linear probe (6–14 MHz). Examinations were made by the same person throughout the study. The hair on the scrotum was shaved and a copious amount of ultrasonic gel was used. Testicular blood flow was monitored in the supratesticular artery (STA) and marginal testicular arteries (MA). After vascular structures were identified and the largest longitudinal portion of the STA and the MA were determined (Figure 1C,D), a color Doppler ultrasound was used to assess TBF, as described for dogs and goats [4,12]. The testicular artery was visualized as blue or red areas representing blood flow towards and away from the probe. The Doppler spectrum cursor was positioned within the lumen of the testicular artery, introducing the pulse wave spectral Doppler. A fixed gate at 2 mm was maintained, the insonation angle was set at 0°, and the baseline was lowered. The angle formed between the Doppler beam and the longitudinal axis of each vessel was ≤60°, while maintaining a pulse repetition frequency of 2000 Hz. In detail, the yielding peak systolic velocity (PSV, cm/s), the end diastolic velocity (EDV, cm/s), the pulsatility index (PI = [SPV − EDV]/mean velocity) and resistive index (RI = [PSV − EDV]/PSV) were computed by the machine based on the specified formula.

### 2.5. Semen Collection and Measurements

Semen collection was conducted following Doppler ultrasound assessment on all days of the experiment. The rams were placed in separate areas away from the other study animals during the collection process, using a specially designed electroejaculator. The probe was approximately 2.5 cm in diameter and 25 cm in length and had three longitudinal electrodes connected to a power source with voltage control. The penis and the inside of the prepuce were washed with physiological saline. We flushed some saline into the prepuce, massaged the prepuce, allowed the saline to flow out, and then dried the preputial opening. The probe and anal sphincter were lubricated, the probe inserted rectally, and the electrostimulation started. The electrostimulation consisted of applying similar voltages for five to ten seconds followed by rest for ten seconds. The starting voltage was 4 V, and it was increased up to a maximum of 8 V. Specially designed semen collection goblets were used to collect ejaculates, which were quickly transferred to the laboratory at 37 °C. We individually collected six ejaculates per animal and subsequently assessed each semen sample separately.

Sperm motility, detailed movement characteristics, and semen concentration were assessed using the Computer-Aided Sperm Analyzer (CASA) system (SCA^®^, Microptic, Barcelona, Spain) in a minimum of five microscope fields or by examining no fewer than 500 cells. Additionally, a negative phase contrast microscope (Nikon, Eclipse, Tokyo, Japan) equipped with a 100× objective lens and a heating plate set at 37 °C was employed for these evaluations. The experiments involved the measurement of various motility parameters, including sperm cell total motility percentage (MS, %), progressive motility percentage (PS, %), curvilinear velocity (VCL, μm/s), straight line velocity (VSL, μm/s), average path velocity (VAP, μm/s), sperm linearity (LIN, calculated as VSL/VCL to represent curvilinear path), straightness (STR, indicated by VSL/VAP to signify the linearity of the average path), and wobble (WOB, a measure of oscillation about the average path expressed as VAP/VCL).

Also, the morphology of spermatozoa was evaluated using the CASA. Morphological evaluation was made with a SpermBlue^®^ kit (Microptic, Barcelona, Spain) and a microscope (Nikon, Eclipse, Tokyo, Japan) with a 40× objective and a blue filter. For that, 15 μL of 25 × 106/mL semen was placed on a glass slide, and a coverslip was held at an angle of about 45° and pushed over the slide to make the smear. The smear was then left to air dry. The dried smear was placed vertically in a jar containing SpermBlue^®^ (Microptic, Spain) fixative for two minutes. The slide was then left to dry at a steepness of 60° to 80°. After the drying process, the smear was dipped twice for three seconds into a jar containing distilled water. The slide was allowed to dry and then stained. After the staining process, the morphology module was selected in the CASA, and evaluations were made on at least 100 spermatozoa. Head, acrosome, tail, and midpiece abnormalities were evaluated and recorded.

The viability of spermatozoa was examined in a smear stained with eosin–nigrosin. For that, 15 µL of eosin–nigrosin dye and 10 µL of semen were placed next to each other on a glass slide and gently mixed with a pipette tip. A coverslip held at an angle of 30° to 40° was used to smear the sample across the slide. The slide was placed on a heating plate until dry, and 200 sperm were evaluated under the light microscope (1000× magnification, oil immersion). Sperm cells stained partially or completely pink or red were considered dead, whereas spermatozoa that did not take the dye into the cell were considered viable.

The functional integrity of the sperm plasma membrane was evaluated using the hypoosmotic swelling (HOS) test developed for human sperm by Jeyendran et al. [13]. The effectiveness of the HOS test was validated prior to testing using a solution of fructose and sodium citrate with an osmolarity of 150 mOsm/kg (hypoosmotic solution (150 mOsm/L) = 7.35 g sodium citrate + 13.51 g fructose prepared in 1 L). We incubated the tube containing 1.0 mL of hypoosmotic solution with 0.1 mL of thoroughly mixed extended semen for one hour at 37 °C. Then, 10 µL of the mixture was placed on a slide and a coverslip was applied. It was immediately evaluated under a phase contrast microscope at 400× magnification. Two hundred spermatozoa were counted in at least five different microscopic fields and the percentage of spermatozoa with swollen and curved tails was recorded.

Total antioxidant (TAS) and total oxidant status (TOS) ELISA test kits (Reel Assay Diagnostics, Gaziantep, Türkiye) were used to determine oxidative and antioxidative capacity (Multiskan GO, Thermo Scientific, Vantaa, Finland). Following sperm collection and dilution, the samples were centrifuged at 800× *g* for ten minutes [14]. After centrifugation, the supernatant was transferred to Eppendorf tubes and kept in a freezer at −20 °C until all samples were prepared. Subsequently, the analysis was performed in accordance with the manufacturer’s protocol.

### 2.6. Statistical Analysis

Data were subjected to a test of normality using the Kolmogorov–Smirnov test to determine homogeneity and data type. In this study, no significant difference was found between the Doppler ultrasound parameters of the studied rams between the right and left testes, and data for each ram were therefore pooled and comparisons were made between the two groups. Data for testicular volume, color Doppler ultrasound parameters, and sperm parameters are presented as mean (MEAN) and standard error of the mean (SEM), and the effect of melatonin on these parameters was tested using the independent sample t-test. The data were analyzed using the Statistical Package for Social Sciences SPSS^®^ (SPSS Inc., Version 26.0, Chicago, IL, USA), and *p*-value < 0.05 was considered significant. Repeated measures were analyzed using a two-way ANOVA to examine the effect of treatment as a fixed factor and time as a repeating factor.

## 3. Results

### 3.1. Testicular Volume

There were effects of treatment (*p* < 0.05), time (*p* < 0.05), and a treatment × time interaction on testicular volume (TV) (*p* < 0.05). In the MEL group, TV increased between W3 and W6 after melatonin administration and was higher at W6, W9, and W12 compared with the CON group (*p* < 0.05) (Figure 2). TV doubled between W3 and W6 in the MEL group (42.08 ± 5.97 vs. 24.38 ± 5.31) and was about four times higher than in the CON group at W6 (100.14 ± 8.33 vs. 26.42 ± 7.56; *p* < 0.05).

### 3.2. Ultrasonographic Examinations

The TBF values between groups are presented in Table 1 and Table 2. Over time, there was an effect of treatment on the PSV values in STA. At W12, PSV was significantly smaller in the MEL group compared with the CON group: 13.93 ± 0.96 cm/s and 20.26 ± 1.26 cm/s, respectively (*p* < 0.05). Furthermore, in the STA, MEL animals had lower RI and PI values between W3 and W9 compared with CON animals (*p* < 0.05). Similarly, also in the MTA, the PI and RI values decreased and were lower between W3 and W9 in the MEL group compared with CON group (*p* < 0.05). Other parameters for color Doppler ultrasound of the testicular arteries showed non-significant differences between the two groups during the study.

### 3.3. Semen Quality

Table 3 summarizes the semen parameters between the MEL and CON groups. There was a positive effect of melatonin (*p* < 0.05) on the percentages for plasma integrity at W3 and W12, and sperm viability at W3 (*p* < 0.05). Table 4 summarizes the motility kinematic parameters between MEL and CON groups. Melatonin exhibited a positive effect on STR, LIN, and WOB at W3, and on VSL, STR, and LIN at W12 (*p* < 0.05). As shown in Table 5, the MEL group exhibited a notably higher percentage of total morphological integrity compared with the CON group. The lowest total abnormal spermatozoa proportion of 5.2 ± 1.7% was observed in the MEL group, whereas the highest proportion of 10% ± 0.5% was recorded in the CON group (*p* < 0.005). The findings from Table 5 suggest that melatonin potentially influences abnormalities in sperm head, tail, and overall morphology over time (*p* < 0.05). Further, there was an increased TAS level at W12 in the MEL group compared with the CON group (*p* < 0.05) (Figure 3). However, there was no difference in sperm concentration, sperm volume, progressive motility, or TOS between the two groups (*p* > 0.05).

## 4. Discussion

Our objective was to investigate the effect of giving a double dose of melatonin to rams during the non-breeding season on testicular volume, testicular blood flow, and spermatological parameters. In this study, higher testicular volume was detected, especially from the third week in the MEL group, with the effect of melatonin applied after the end of the season. Moreover, there was a decrease in the PI and RI values starting from the third week, which define high blood flow in the MEL group compared with the CON group. These findings supported our hypothesis that melatonin significantly improved testicular blood flow, testicular volume, and oxidant status. The magnitude of the response to treatment, the seasonal period during melatonin treatment, and the intensity of the breeding program for the rams seem to be effective.

In this study, the mean testicular volume decreased in the CON group, which coincided with the end of the breeding season. It increased in the MEL group, especially between the third and sixth week. A second increase in testicular volume was recorded between the ninth and twelfth week, which was about five to eight weeks after the application of the second dose of melatonin. The accurate determination of testicular volume is of great potential in the evaluation of testicular function. Testicular volume largely reflects spermatogenesis because the seminiferous tubules make up 70–80% of the testicular mass [15,16]. While testicular volume was significantly higher (*p* < 0.05) from spring to summer in Racka [17], Karakul [18], and Suffolk rams [19], lower values were measured in autumn and winter. In a study with fat-tailed rams [20], the lowest and highest values for testicular volume were recorded in autumn (October) and early winter (December), respectively. The standard ovine breeding season in the northern hemisphere is from September to December, with a successive lambing period from February to May the following year [21]. The northern hemisphere rams used in this study were physically mature. Therefore, changes in testicular volume can be attributed to the effect of melatonin. The increase in testicular volume six weeks after implantation suggests that the testicular parenchyma proliferated the same as during the natural breeding season. Lincoln et al. [22] and Casao et al. [23] showed the ability of melatonin treatment to downregulate prolactin, which promotes follicle-stimulating hormone (FSH) and luteinizing hormone (LH) upregulation. All these factors influence the Sertoli cells’ sensitivity to FSH during the release of exogenous melatonin and regulate cellular growth, proliferation, and the number of different testicular cell types [24]. Upregulation of these endocrine factors probably contributes to increased testicular volume.

In this study, RI and PI indexes decreased in the rams receiving melatonin compared with the CON group, especially from the third week onwards after melatonin administration. The resistance (RI) and pulsatility (PI) indexes determined with color Doppler ultrasound provide more reliable information about testicular vasculature and blood flow velocity [25]. Melatonin can modulate testicular blood flow and semen quality by regulating different levels of the hypothalamic–pituitary–gonadal axis in seasonally breeding mammals [26]. The different vascular effects observed with melatonin are attributed to the relative distribution of the melatonin receptors M1 and M2 [27]. M1 and M2 receptors exist in the ram reproductive tract [28]. For this reason, it has been possible to monitor an increase in blood flow in the testis with Doppler ultrasonography, with the thought that there may be an increase in the number of M1 and M2 receptors because of melatonin hormone administered during the non-breeding season. Decreased values for RI and PI indicate decreased blood flow resistance, which results in an increased testicular perfusion and an improved supply with oxygen and nutrients to the testis [9]. Salama et al. [29] and Samir et al. [30] showed that RI and PI in the supratesticular arteries have a robust negative correlation with blood velocities parameters (PSV and EDV), testicular volume, and intratesticular colored areas and, therefore, showed a significant improvement in testicular blood flow because of the melatonin administration. Hedia et al. [20] correlated a significant decrease in testicular volume with an increase in RI and PI, i.e., with a decrease in testicular blood flow during warm months. These results align with the results of this study, as we observed an increase in RI and PI values in the CON group outside the breeding season. Furthermore, testicular RI and PI indices have been shown to be highly correlated with sperm quality parameters in dogs [31] and in detecting sexual dysfunction disorders in humans [25].

Our results are in line with the study conducted in Ossimi rams [32] that received melatonin and showed a similar decrease in RI and PI values from around the fourth week onwards. Seasonal variation causes this slight difference in the onset of melatonin action. However, this study showed that melatonin did not affect sperm motility or morphology throughout the entire study. However, melatonin-treated rams showed normal cell development in the testis in the season. Previous studies on melatonin administration in rams during the non-breeding season gave mixed results. For instance, Ramadan et al. [5] reported that in Damascus bucks the beneficial effect of photoperiodic treatment and melatonin application on semen properties was not as prominent during the non-breeding season. Melatonin administration did not improve semen parameters in black Racka rams even out of season [2]. Contrarily, melatonin administration had a positive effect on semen quality in buffaloes [33], bucks [34], and rams [32] during the non-breeding season. According to Casao et al. [35], melatonin exhibits a two-phase impact on the male reproductive system. Its short-term influence relates to progressive sperm motility, while medium-term effects involve alterations in kinematic motility parameters. The notably higher VCL and lower VAP observed in spermatozoa suggest significant midpiece bending and extensive lateral head displacement. The discrepancies noted, specifically in the STR and LIN values obtained by the twelfth week, imply that melatonin’s effects might manifest in the long term concerning alterations in VCL and VAP values. On the other hand, Casao et al. [35] observed an effect of melatonin implants on sperm kinematic parameters in non-breeding season rams, and this effect appeared approximately 91–105 days after implantation. Thus, one reason we failed to observe an effect of the treatment could be that the duration of our study was shorter than that of Casao et al. [35]. We observed an increase in testicular volume approximately three weeks after the first and second melatonin implants, respectively. The increase in testicular development will make a difference in the quality of potential new spermatozoa. When the spermatogenesis length for the rams is considered [36], if size increased six weeks after implant and it takes seven to nine weeks for sperm production, maybe we would have seen an effect on sperm characteristics had we taken one more semen sample at week 16 after the implant.

Melatonin implants applied to two different breeds of ram exposed to a natural increase in daylength, without any artificial light treatment prior to implantation, did not affect semen volume or concentration [37]. Those results are consistent with those of this study. For the rams to respond to short days or alternative melatonin therapy, they need to be exposed to long days for a long time (one and a half to two months), and it has been determined that resistance to sexually stimulating short photoperiods can be achieved [5,26,38]. The effect of exogenous melatonin on reproductive endocrinology, semen quality and production, testicular size, and libido was investigated in Merino and Poll Dorset rams, and no changes in sperm motility or morphology were recorded in response to treatment [39]. The underlying basis for the mentioned variances remains unclear. Nonetheless, there seems to be a consistency in the circadian rhythm of melatonin levels across diverse breeds. Factors potentially influencing this rhythm could include breed-specific attributes, seasonal variations, geographical location, procedural differences in handling semen, and variations in equipment utilized. Rams tend to maintain high sperm parameters for some time after the breeding season is over, and no sudden change in these parameters is observed. This lack of improvement in overall semen quality could be explained by the fact that many rams in the MEL group had high-quality ejaculates prior to melatonin implantation and were actively used in sheep breeding, although more investigation is needed before making such an assumption. Although not part of this study, Rekik et al. [40] reported that melatonin did not cause changes in semen characteristics of young Barbarine rams, although melatonin treatment improved mating behaviors like increased total activity time and the number of lateral approaches and mount trials.

In this study, no difference was recorded between the treatment groups regarding the total oxidant status (TOS), whereas the total antioxidant status (TAS) was higher in the MEL group at week twelve. The critical role of melatonin as an antioxidant may potentially be in pathophysiological conditions subject to oxidative stress [41]. The animals in this study were not under such stressful conditions as to evoke this important role of melatonin. Melatonin can increase membrane integrity by shielding the mitochondria and plasma membrane from free radicals, lipid peroxides, and reactive oxygen species [42]. Therefore, the significant increase in TAS at week twelve can be correlated with the increase in plasma membrane integrity at week twelve.

## 5. Conclusions

Melatonin’s exogenous administration during the non-breeding season improves testicular blood flow by lowering the RI and PI of testicular arteries, as determined by color Doppler ultrasound. The current study’s findings lead to at least three certain conclusions. First, the blood flow to the testis of treated rams increased, and as a result, it is possible to observe improvements in the endogenous and exogenous function of the testis. Secondly, the findings suggest that melatonin administration could help to lengthen the breeding season. Thirdly, melatonin, which causes an increase in testicular blood flow, was insufficient to make a difference in sperm parameters in the MEL group compared with the CON group. Moreover, it should be noted that there is a need for further studies in which large numbers of animals are used, animals are investigated for longer than twelve weeks after hormone implantation, different hormonal treatments are applied, and stimulations are used.

## Figures and Tables

**Figure 1 animals-14-00442-f001:**
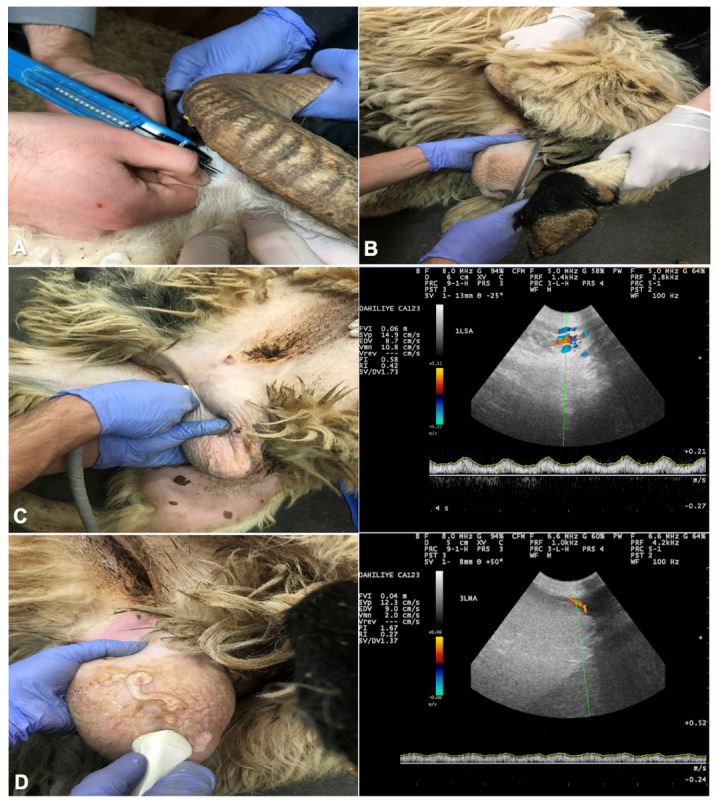
The site of subcutaneous injection of the melatonin implant at base of the ear (**A**); measurement of testicular length with a digital caliper to calculate testicular volume (**B**); assessment of the blood flow within the supratesticular artery (**C**) and marginal testicular artery (**D**) with color Doppler ultrasound.

**Figure 2 animals-14-00442-f002:**
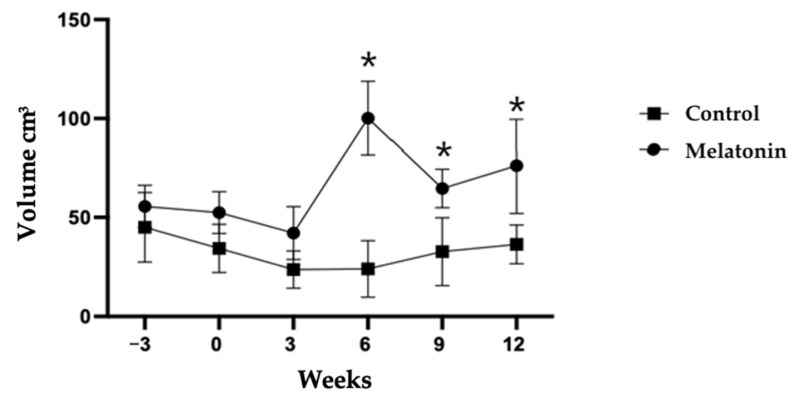
Testicular volume (mean ± SEM) determined for rams treated (MEL) or not (CON) with melatonin. * *p* < 0.05 between treatment groups.

**Figure 3 animals-14-00442-f003:**
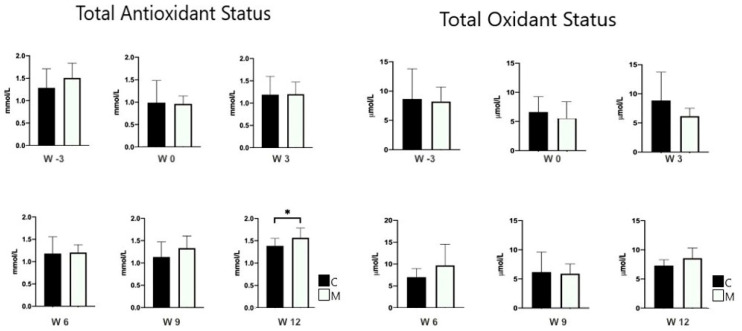
Total oxidant and antioxidant status (mean ± SEM) determined in semen of rams treated (MEL) or not (CON) with melatonin. * *p* < 0.05 between treatment groups.

**Table 1 animals-14-00442-t001:** Testicular blood flow parameters (mean ± SEM) in the supratesticular artery measured with color Doppler ultrasound in rams treated (MEL) or not (CON) with melatonin.

WEEKS	PSV (cm/s)		EDV (cm/s)		PI		RI	
	CON	MEL	CON	MEL	CON	MEL	CON	MEL
**W-3**	19.54 ± 2.00	17.39 ± 2.15	8.67 ± 0.54	7.72 ± 0.51	0.78 ± 0.06	0.89 ± 0.12	0.53 ± 0.02	0.53 ± 0.04
**W0**	14.64 ± 1.47	18.46 ± 1.89	8.03 ± 0.55	8.00 ± 0.52	0.71 ± 0.04	0.91 ± 0.09	0.59 ± 0.01	0.50 ± 0.05
**W3**	13.28 ± 0.95	16.00 ± 1.64	7.39 ± 0.57	8.29 ± 0.54	0.65 ± 0.03	0.80 ± 0.12 *	0.45 ± 0.01	0.48 ± 0.06 *
**W6**	14.73 ± 0.88	15.00 ± 0.82	8.30 ± 0.60	7.10 ± 0.42	0.65 ± 0.05	0.84 ± 0.05 *	0.43 ± 0.02	0.41 ± 0.03 *
**W9**	15.13 ± 1.23	13.56 ± 0.94	7.80 ± 0.59	8.37 ± 0.58	0.74 ± 0.08	0.55 ± 0.03 *	0.51 ± 0.03	0.38 ± 0.01 *
**W12**	20.26 ± 1.26	13.93 ± 0.96 *	8.81 ± 1.08	7.08 ± 0.58	0.82 ± 0.06	0.71 ± 0.04	0.56 ± 0.04	0.49 ± 0.02

* *p* < 0.05 between treatment groups. PSV = peak systolic velocity (cm/s); EDV = end diastolic velocity (cm/s); RI = resistance index; PI = pulsatility index.

**Table 2 animals-14-00442-t002:** Testicular blood flow parameters (mean ± SEM) in the marginal artery measured with color Doppler ultrasound in rams treated (MEL) or not (CON) with melatonin.

WEEKS	PSV (cm/s)		EDV (cm/s)		PI		RI	
	CON	MEL	CON	MEL	CON	MEL	CON	MEL
**W-3**	14.04 ± 0.52	11.83 ± 0.80	8.84 ± 0.45	8.03 ± 0.65	0.54 ± 0.04	0.50 ± 0.08	0.37 ± 0.03	0.35 ± 0.05
**W0**	12.21 ± 0.54	11.53 ± 1.12	8.08 ± 0.24	7.50 ± 0.69	0.47 ± 0.05	0.55 ± 0.07	0.33 ± 0.03	0.38 ± 0.05
**W3**	10.39 ± 0.56	11.24 ± 1.45	7.33 ± 0.47	6.98 ± 0.74	0.41 ± 0.07	0.61 ± 0.06 *	0.29 ± 0.03	0.41 ± 0.05
**W6**	11.26 ± 0.76	11.40 ± 0.76	7.08 ± 0.60	8.25 ± 0.56 *	0.54 ± 0.07	0.39 ± 0.05 *	0.35 ± 0.04	0.28 ± 0.02 *
**W9**	10.51 ± 0.59	10.34 ± 0.80	7.12 ± 0.45	7.36 ± 0.66	0.47 ± 0.03	0.34 ± 0.04 *	0.35 ± 0.03	0.28 ± 0.03
**W12**	13.17 ± 0.94	12.47 ± 0.92	8.26 ± 0.83	8.04 ± 0.69	0.54 ± 0.03	0.48 ± 0.04	0.38 ± 0.02	0.35 ± 0.01

* *p* < 0.05 between treatment groups. PSV = peak systolic velocity (cm/s); EDV = end diastolic velocity (cm/s); RI = resistance index; PI = pulsatility index.

**Table 3 animals-14-00442-t003:** Spermatological parameters (mean ± SEM) of rams treated (MEL) or not (CON) with melatonin.

WEEKS	Volume (mL)		Total Motility (%)		Progressive Motility (%)		Concentration (×10^9^ Sperm/mL)		Viability (%)		HOST (%)	
	CON	MEL	CON	MEL	CON	MEL	CON	MEL	CON	MEL	CON	MEL
**W-3**	0.73 ± 0.10	0.52 ± 0.10	93.75 ± 3.50	93.88 ± 1.30	73.33 ± 10.10	70.74 ± 3.90	1.37 ± 0.70	0.99 ± 0.80	72.00 ± 2.90	76.40 ± 1.80	69.50 ± 2.10	75.80 ± 1.60
**W0**	0.63 ± 0.10	0.49 ± 0.20	90.56 ± 0.90	92.95 ± 2.30	72.76 ± 4.40	71.43 ± 2.80	1.18 ± 0.30	1.07 ± 0.40	80.10 ± 0.90	81.00 ± 1.50	77.25 ± 1.70	79.36 ± 1.10
**W3**	0.81 ± 0.60	0.60 ± 0.20	97.48 ± 0.60	96.28 ± 0.90	85.83 ± 3.20	77.81 ± 5.20	1.3 ± 0.20	1.32 ± 0.80	87.50 ± 1.40	84.00 ± 1.30 *	80.40 ± 0.90	82.50 ± 0.90 *
**W6**	0.80 ± 0.10	1.09 ± 0.20	97.67 ± 1.10	96.25 ± 2.40	84.56 ± 1.50	80.83 ± 6.70	1.36 ± 0.50	1.31 ± 0.20	84.75 ± 1.30	81.60 ± 0.60	82.75 ± 2.10	78.20 ± 2.80
**W9**	0.90 ± 0.10	1.20 ± 0.20	93.89 ± 2.40	97.07 ± 0.80	77.00 ± 6.10	82.44 ± 2.60	1.10 ± 0.30	1.12 ± 0.40	78.00 ± 1.50	78.40 ± 1.20	77.50 ± 0.90	79.60 ± 1.60
**W12**	0.93 ± 0.20	1.54 ± 0.50	92.23 ± 1.10	95.86 ± 1.40	74.47 ± 5.50	85.96 ± 2.40	1 ± 0.50	0.81 ± 0.10	78.25 ± 1.10	82.80 ± 1.80 *	74.00 ± 1.90	82.80 ± 1.80 *

* *p* < 0.05 between treatment groups.

**Table 4 animals-14-00442-t004:** Sperm kinematic parameters (mean ± SEM) of rams treated (MEL) or not (CON) with melatonin.

WEEKS	VCL (μm/s)		VAP (μm/s)		VSL (μm/s)		STR (μm/s)		LIN (μm/s)		WOB (μm/s)	
	CON	MEL	CON	MEL	CON	MEL	CON	MEL	CON	MEL	CON	MEL
**W-3**	80.37 ± 11.20	92.16 ± 10.05	50.08 ± 8.01	67.06 ± 7.14	31.83 ± 6.46	45.33 ± 4.50	58.69 ± 2.49	64.26 ± 2.15	38.49 ± 3.05	48.12 ± 2.59	61.59 ± 1.98	70.55 ± 1.66
**W0**	105.22 ± 14.02	107.99 ± 8.64	60.92 ± 9.70	74.99 ± 5.75	36.91 ± 7.59	49.93 ± 4.51	55.85 ± 2.48	63.97 ± 2.56	34.2 ± 2.46	46.78 ± 3.56	57.98 ± 1.80	68.70 ± 2.56
**W3**	130.27 ± 16.85	123.83 ± 7.23	71.76 ± 11.39	82.93 ± 4.37	42.00 ± 8.72	54.53 ± 4.53	53.02 ± 2.47	63.69 ± 2.98 *	29.92 ± 1.87	45.44 ± 4.54 *	54.37 ± 1.63	66.86 ± 3.46 *
**W6**	92.68 ± 10.38	104.47 ± 11.28	67.26 ± 8.17	69.35 ± 6.92	47.80 ± 6.06	47.58 ± 6.40	67.60 ± 2.32	63.13 ± 4.52	51.47 ± 3.21	44.73 ± 6.46	71.26 ± 1.79	65.14 ± 5.03
**W9**	97.96 ± 6.65	87.68 ± 8.90	75.75 ± 6.03	62.31 ± 9.43	58.91 ± 4.30	46.32 ± 8.65	69.78 ± 0.83	66.67 ± 3.80	54.40 ± 1.39	48.64 ± 5.05	72.94 ± 1.57	67.86 ± 4.07
**W12**	106.45 ± 17.59	115.21 ± 1.85	81.74 ± 14.64	94.18 ± 5.01	76.00 ± 17.14	57.15 ± 8.57 *	64.44 ± 1.48	76.54 ± 4.79 *	49.19 ± 0.86	64.31 ± 6.83 *	71.81 ± 1.49	65.33 ± 15.18

* = *p* < 0.05 between treatment groups. VCL = curvilinear velocity; VAP = average path velocity (μm/s); VSL = straight line velocity (μm/s); STR = straight line rate; LIN = sperm linearity (curvilinear path); WOB = wobble.

**Table 5 animals-14-00442-t005:** Abnormal spermatozoa (mean ± SEM) of rams treated (MEL) or not (CON) with melatonin.

WEEKS	Head (%)		Midpiece (%)		Tail (%)		Total (%)	
	Con	Mel	Con	Mel	Con	Mel	Con	Mel
**W-3**	2.2 ± 0.2	1.5 ± 0.3	1.6 ± 0.8	1.2 ± 0.2	6.4 ± 0.3	3.8 ± 0.7	10.0 ± 0.5	7.2 ± 1.1
**W0**	2.3 ± 0.5	1.6 ± 0.8	0.9 ± 0.2	0.7 ± 0.3	4.7 ± 0.7	3.9 ± 1.1	7.5 ± 1.3	6.2 ± 0.5
**W3**	2.4 ± 0.3	1.5 ± 0.7	1.4 ± 0.3	0.8 ± 0.7	5.6 ± 0.8	4.8 ± 0.7	9.0 ± 1.1	7.0 ± 1.4
**W6**	2.2 ± 0.7	1.6 ± 0.3	1.2 ± 0.2	0.6 ± 0.3	5.8 ± 1.7	5.0 ± 0.5	7.6 ± 2.3	7.2 ± 1.7
**W9**	2.2 ± 0.7	1.4 ± 0.3	0.9 ± 0.2	1.0 ± 0.3	6.2 ± 1.7	4.3 ± 0.8 **	8.8 ± 4.7	6.2 ± 6.2 *
**W12**	2.2 ± 0.7	1.1 ± 0.5 *	1.1 ± 0.5	0.6 ± 0.8	6.8 ± 0.7	4.0 ± 0.5 **	9.8 ± 1.7	5.2 ± 1.7 **

* *p* < 0.05 between treatment groups. ** *p* < 0.005 between treatment groups.

## Data Availability

The data presented in this study are available on request from the corresponding author. The data are not publicly available due to the original contributions presented in the study are included in the article, further inquiries can be directed to the corresponding author.

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
