# Peer review of "Melatonin Administration Enhances Testicular Volume, Testicular Blood Flow, Semen Parameters and Antioxidant Status during the Non-Breeding Season in Bafra Rams"

_animals, 2024, doi:10.3390/ani14030442_

Round 1
Reviewer 1 Report
Comments and Suggestions for Authors
The manuscript aimed to investigate the impact of melatonin administration on testicular volume, testicular blood flow, semen parameters, and antioxidant status during the non-breeding season in Bafra rams.
The authors made a good attempt to explore the role of melatonin implants in improving the mentioned reproductive parameters of Bafra rams; however, there were few shortcomings in the experimental methodology, parameter observed, and the results. 1) How many ejaculates from each ram were collected after each ultrasound examination. 2) Were semen samples from rams pooled? or evaluated individually for total antioxidant and oxidant status, 3) Analysis of semen morphology and CASA motility parameters must be provided in detail. It should be clearly described what different morphological abnormalities were counted and what were their ratio in semen samples before or after the melatonin treatment, 4) Authors must provide the details of motility kinematics apart from total and progressive motility the results section (4) Please provide description how doppler images were acquired. Details of setting up of region of interest and angle of incidence for doppler must be included, 5) Reviewer suggest that authors should have included important hormones to provide insight about the possible impact of melatonin. These hormones include FSH, LH, melatonin, and testosterone. Also, authors did not provide any data on the libido scoring of the bucks when exposed to females after melatonin treatment.
Comments on the Quality of English LanguageOverall, the manuscript is well-written and there are minor issues with the use of the English language.
Author Response
1) We would like to express our gratitude to the reviewer for their valuable comment. Semen collection was conducted following Doppler ultrasound assessment on all experimental days. The rams were placed in separate areas away from the other study animals during the collection process, using a specially designed electro-ejaculator.
2)We would like to express our gratitude to the reviewer for their valuable comment. The semen parameters and antioxidant status were assessed individually to thoroughly analyze their respective impacts on the reproductive system of the rams. We collected 6 ejaculates per animal individually, then assessed all semen samples individually.
3)We extend our sincere appreciation to the reviewer for their invaluable feedback and insightful comments. Hereby, we presented more detail for motility assesment. Sperm motility, detailed movement characteristics, and semen concentration were assessed using the Computer Aided Sperm Analyzer (CASA) system (SCA®, Microptic, Barcelona, Spain) in a minimum of 5 microscope fields or by examining no fewer than 500 cells. Additionally, a negative phase-contrast microscope (Nikon, Eclipse, Tokyo, Japan) equipped with a 100x objective lens and a heating plate set at 37°C was employed for these evaluations.The experiments involved the measurement of various motility parameters, including sperm cell total motility percentage (MS, %), progressive motility percentage (PS, %), curvilinear velocity (VCL, μm/s), straight line velocity (VSL, μm/s), average path velocity (VAP, μm/s), sperm linearity (LIN, calculated as VSL/VCL to represent curvilinear path), straightness (STR, indicated by VSL/VAP to signify the linearity of the average path), wobble (WOB, a measure of oscillation about the average path expressed as VAP/VCL), amplitude of lateral sperm head displacement (ALH, μm), and beat cross frequency (BCF, Hz).
4) We extend our sincere appreciation to the reviewer for their invaluable feedback and insightful comments. We have presented a table illustrating sperm kinematic parameters.
Hereby more detail description refarding doppler images. The Doppler spectrum cursor was positioned within the lumen of the testicular artery, introducing the pulse wave spectral Doppler. A fixed gate at 2 mm was maintained, the insonation angle was set at 0°, and the baseline was lowered. The angle formed between the Doppler beam and the longitudinal axis of each vessel was ≤60 degrees, while maintaining a pulse-repetition frequency of 2,000 Hz. In detail, the yielding peak systolic velocity (PSV, cm/second), the end-diastolic velocity (EDV, cm/second), the pulsatility index (PI = [SPV−EDV]/mean velocity) and resistive index RI = [PSV−EDV]/PSV) were computed by the machine based on the specified formula.)
5)We would like to express our gratitude to the reviewer for their valuable comment. In the project content, we haven't included hormone profile analyses. I'm concerned that we won't be able to provide any data associated with the hormones suggested by the reviewers. This project is planned to be carried out during the non-breeding season. Consequently, there were no ewes displaying estrus behavior. Therefore, we couldn't include the libido scoring of the rams by using the females.
Thank you for your insightful comments and guidance.
Reviewer 2 Report
Comments and Suggestions for Authors
In the present work, Akar et al. try to verify that melatonin administration enhances testicular volume, testicular blood flow, semen parameters, and antioxidant status during the non-breeding season in Bafra rams. However, some questions also should be explained.
1. In fact, melatonin administration has favourable effects on testicular size and function and reproductive capacity during the non-breeding season. However, what is the aim for this study? As the authors stated that the objectives were to determine effects of exogenous melatonin in rams during the non-breeding season on TBF, testicular volume, and semen quality and oxidant status. The semen with high quality was used to artificial insemination? In general, ewes have no oestrous behaviour during the non-breeding season.
2. In general, rams have low semen quality during the non-breeding season. However, in this paper, semen quality of rams during the non-breeding season was not low for CON (72±2.9 %, Table 3), which can be used for artificial insemination.
3. Table 3, please explain that the significant differences between Volumes for CON and MEL at W -3 (0.73±0.1 VS 0.52±0.1), W3 (0.81±0.6 VS 0.6±0.2), as well as differences between Volumes for CON at different stages W -3 (0.73±0.1), W3 (0.81±0.6), and W 12 (0.93±0.2). These differences are not acceptable.
4. The data in Table 3 are too crowded.
5. What is the novelty for this paper ? Melatonin administration enhances testicular volume, testicular blood flow, semen parameters during the non-breeding season in rams, which have been reported by El-Shalofy et al. (2022). Melatonin administration enhances antioxidant status of sperm, which have been reviewed by Makris et al. (2023). This paper is not cited this manuscript.
Makris A, Alevra AI, Exadactylos A, Papadopoulos S. The Role of Melatonin to Ameliorate Oxidative Stress in Sperm Cells. Int J Mol Sci. 2023;24(20):15056.
Comments on the Quality of English LanguageModerate editing of English language required.
Author Response
1) We would like to express our gratitude to the reviewer for their valuable comment. In this paper, our plan did not involve using high-quality semen for artificial insemination. Instead, our aim was to consolidate data concerning the effects of melatonin on testicular blood flow and its antioxidant properties.
2) We would like to express our gratitude to the reviewer for their valuable comment. Previous literature reviews within the scope of our study align with our current motility results. In this context, there doesn't appear to be a noticeable decrease in motility evaluation during our off-season compared to the breeding season in Control group.
3) We would like to express our gratitude to the reviewer for their valuable comment. We didn't demonstrate any difference in melatonin's effect on semen volume. Moreover, it's possible that the rams became accustomed to the electroejaculator, which could explain the observed increase in sperm volume in the control group.
4)We would like to express our gratitude to the reviewer for their valuable comment. We have removed one parameter from Table 4 in an effort to make it more understandable.
5)We would like to express our gratitude to the reviewer for their valuable comment. The novelty of this paper lies in combining the effects of melatonin on blood flow regulation and its antioxidant properties.
Thank you for your insightful comments and guidance.
Reviewer 3 Report
Comments and Suggestions for Authors
This manuscript deals with the effects of treating rams of a seasonal breed (Bafra) with exogenous melatonin in the out of season upon a lot of characteristics related to their reproductive performance, i.e., testicular volume, testicular blood flow and sperm characteristics (morphology, viability, functional integrity and oxidant and antioxidant status of spermatozoa). This is a very important topic in sheep reproduction mainly in higher latitudes because of the reproductive seasonality of this species and the importance that rams have in the reproductive success in the flock. Results have shown some effects of melatonin treatment - for instance, an increase of testicular volume, which has not clearly been seen in other studies in which the melatonin was applied in out of season, unless the rams had experienced a previous period of exposure to long days. However, no positive effect was found in what concern to semen volume, sperm motility, sperm concentration and total oxidant and antioxidant status of semen.
The manuscript is well written. The experiment was well designed and conducted and results are well discussed with relevant literature presented. The findings might be of interest to readers of Animals, particularly of the Animal Reproduction section.
I make the following comments that in my opinion the authors should consider in any revision of the manuscript:
1. Page 2, line 80. “Capra hircus” is the scientific name for domestic goat and “Ovis aries” is the scientific name for domestic sheep. Correct accordingly.
2. Page 5, line 146. Delete “(SCA®, Mi-146 croptic, Barcelona, Spain)” as this information for CASA has already been given before.
3. In Statistical Analysis, section 2.6., I don´t understand why the authors used the paired sample t-test to test the effect of melatonin on testicular volume and sperm parameters. The comparison between the two groups (melatonin treated and control) can be compared statistically at a precise time using the independent sample t-test or one way ANOVA (the result is the same). The factorial repeated measures ANOVA should be used if it is to include both the treatments (fator) and time (weeks - repeated measures) in the model, i.e., treatments as between subject factor and time (weeks) as within-subject variable. I feel that this is not well explained in the text.
4. In Figure 2., don´t the authors feel that the difference of 400% in testis volume in week 6 is too high? Have the authors checked well the data for outliers? This graph lacks the unit of testicular volume (cm3).
5. Uniformize the designation of treatments along the Figures and Tables.
6. Table 1. Replace the Title by “Testicular blood flow parameters (mean ± SEM) in the supratesticular artery measured with Color Doppler ultrasound in rams treated (Mel) or not (Con) with melatonin.”
7. Table 2. Replace the Title by “Testicular blood flow parameters (mean ± SEM) in the marginal artery measured with Color Doppler ultrasound in rams treated (Mel) or not (Con) with melatonin.”
8. Table 2. Replace the Title by “Spermatological parameters (mean ± SEM) of rams treated (Mel) or not (Con) with melatonin.”
9. Page 7, line 223 to 225. The statment “There was a positive effect of melatonin (P < 0.05) on the percentages of plasma integrity at W3 and W12” is not consistent with the results of Table 3, as the value for MEL is significantly lower than that for CON in week 3 but significantly higher in week 9.
10. Figure 3. Replace the Title by “Total oxidant and antioxidant status (mean ± SEM) determined in semen of rams treated (Mel) or not (Con) with melatonin.”
11. Page 10, line 332. Replace “improvement” by “improved”.
Author Response
1) We express our gratitude to the reviewer for their valuable comments. The revision has been made based on the suggestions provided by the reviewer.
2) We express our gratitude to the reviewer for their valuable comments. The revision has been made based on the suggestions provided by the reviewer.
3) We express our gratitude to the reviewer for their valuable comments. Statistical analysis part has revised by the project team.
In this regards,
Data were subjected to a test of normality using the Kolmogorov-Smirnov test to determine homogeneity and data type. In this study, no significant difference was found between the Doppler ultrasound parameters of the studied rams between the right and left testes, so that the data for each ram were pooled and comparisons were made between the two groups. Data for testicular volume, Color Doppler ultrasound parameters, and sperm parameters are presented as mean (MEAN) and standard error of the mean (SEM), and the effect of melatonin on these parameters was tested using the independent sample t-test. The data were analysed using the Statistical Package for Social Sciences SPSS® (SPSS Inc., Version 26.0, Chicago, IL, USA), and P-value Ë‚ 0.05 was considered significant. Repeated measures were analyzed using a two-way ANOVA to examine the effect of treatment as a fixed factor and time as a repeating factor.
4) We express our gratitude to the reviewer for their valuable comments. In Figure 2, we analyzed the weekly changes within both the melatonin group and the control groups. Specifically, the observed increase from 49 cm³ to 100 cm³ in testicular volume during the 6th week was, on average, among the animals in the melatonin group.
5) We express our gratitude to the reviewer for their valuable comments. The revision has been made based on the suggestions provided by the reviewer.
6) We express our gratitude to the reviewer for their valuable comments. The revision has been made based on the suggestions provided by the reviewer.
7) We express our gratitude to the reviewer for their valuable comments. The revision has been made based on the suggestions provided by the reviewer.
8) We express our gratitude to the reviewer for their valuable comments. The revision has been made based on the suggestions provided by the reviewer.
9)We express our gratitude to the reviewer for their valuable comments. The statistical analysis results were rechecked, revealing that values had been placed in the wrong sections. Consequently, necessary adjustments were made accordingly.
10) We express our gratitude to the reviewer for their valuable comments. The revision has been made based on the suggestions provided by the reviewer.
Thank you for your insightful comments and guidance.
Round 2
Reviewer 2 Report
Comments and Suggestions for Authors
Thanks for author’s responses. However, some questions also should be explained.
1. In general, rams have low semen quality during the non-breeding season. However, in this paper, the progressive motility (%) during the non-breeding season was 72.76 % (Table 4) for CON. In fact, the progressive motility (%) before the breeding season is 54.1 % in rams (Kozłowska et al., 2022), and the progressive motile spermatozoa of semen collected from rams during non-breeding season is 37-42 % (Casao et al., 2010).
Casao A, Vega S, Palacín I, Pérez-Pe R, Laviña A, Quintín FJ, Sevilla E, Abecia JA, Cebrián-Pérez JA, Forcada F, Muiño-Blanco T. Effects of melatonin implants during non-breeding season on sperm motility and reproductive parameters in Rasa Aragonesa rams. Reprod Domest Anim. 2010;45(3):425-32.
Kozłowska N, Faundez R, Borzyszkowski K, Dąbrowski S, Jasiński T, Domino M. The Relationship between the Testicular Blood Flow and the Semen Parameters of Rams during the Selected Periods of the Breeding and Non-Breeding Seasons. Animals. 2022;12(6):760.
2. Table 4, Volumes for CON and MEL at W -3 are 0.73 and 0.521, and concentrations for CON and MEL are 1.37 and 0.99 at W -3. These differences are not acceptable in experimental design.
Comments on the Quality of English LanguageExtensive editing of English language required.
Author Response
1) We would like to thank the reviewer for the valuable comments. Various studies in disparate literature sources have reported lower levels of progressive motility. However, our investigation aligns with findings from certain studies demonstrating similar levels of progressive motility (e.g. in Greece (Karagiamides et al., 2000) and Iran (Moghaddam et al., 2012). Notably, in the literature review, I encountered studies showcasing comparable progressive motility to our research. Some even displayed higher data in the control group compared to the melatonin treatment group (Egerszegi et al., 2014; Martí et al., 2012).
Moreover, upon reviewing the study forwarded by you (Cassoa et al. (2010)), it was the sample size consisted of only 4 rams (Control=2, melatonin=2), raising concerns regarding its validity. Additionally, the inability to calculate standard deviation (SD) or standard error (SE) in this study poses further limitations. Furthermore, their methodology involved semen dilution before examination. Consequently, outcomes could potentially vary based on factors such as breed, specific timing within the year, geographical location, equipment variations, as well as semen handling and processing techniques. Specifically, within line 402 of our manuscript, we have tried to explain in more detail what we have argued about the results obtained in terms of motility.
While the pattern of melatonin levels seems consistent across different breeds, it is reasonable that tissue sensitivity to melatonin could differ among individual animals.
2) We would like to thank the reviewer for the valuable comments. While a numerical variance exists between the values, most important is that there is no statistical difference. Only highlighting the mean without accounting for the standard error of the mean (SEM) might lead to incomprehensible. For example, examining concentrations such as 1.37 +/- 0.7 and 0.99 +/- 0.8, the inclusion of SEM evidently indicates the lack of difference between the values.